# Wound Healing and Antioxidant Evaluations of Alginate from *Sargassum ilicifolium* and Mangosteen Rind Combination Extracts on Diabetic Mice Model

**Pugar Arga Cristina Wulandari** [1], **Zulfa Nailul Ilmi** [1], **Saikhu Akhmad Husen** [2], **Dwi Winarni** [2], **Mochammad Amin Alamsjah** [3], **Khalijah Awang** [4], **Marco Vastano** [5], **Alessandro Pellis** [5,6], **Duncan MacQuarrie** [5] and **Pratiwi Pudjiastuti** [1,*]

1. Department of Chemistry, Faculty of Science and Technology, Airlangga University, Surabaya 60115, Indonesia; cristinapugar@gmail.com (P.A.C.W.); zulfanailulilmi@gmail.com (Z.N.I.)
2. Department of Biology, Faculty of Science and Technology, Airlangga University, Surabaya 60115, Indonesia; saikhu-a-h@fst.unair.ac.id (S.A.H.); dwi-w@fst.unair.ac.id (D.W.)
3. Department of Marine, Faculty of Fisheries and Marine, Airlangga University, Surabaya 60115, Indonesia; alamsjah@fpk.unair.ac.id
4. Department of Chemistry, Faculty of Science, University of Malaya, Kuala Lumpur 50603, Malaysia; khalijah@um.edu.my
5. Department of Chemistry, University of York, Heslington, York YO10 5DD, UK; marco.vastano@york.ac.uk (M.V.); alessandro.pellis@boku.ac.at (A.P.); duncan.macquarrie@york.ac.uk (D.M.)
6. Department for Agrobiotechnology, IFA-Tulln, Institute for Environmental Biotechnology, University of Natural Resources and Life Sciences, Konrad Lorenz Strasse 20, Tulln an der Donau, 3430 Vienna, Austria
* Correspondence: pratiwi-p@fst.unair.ac.id; Tel.: +62-856-3390-952

**Abstract:** A diabetic foot ulcer is an open wound that can become sore and frequently occurs in diabetic patients. Alginate has the ability to form a hydrophilic gel when in contact with a wound surface in diabetic patients. Xanthones are the main compounds of mangosteen rind and have antibacterial and anti-inflammatory properties. The purpose of this research was to evaluate the wound healing and antioxidants assay with a combination of alginate from *S. ilicifolium* and mangosteen rind combination extracts on a diabetic mice model. The characterization of alginate was carried out by size exclusion chromatography with multiple angle laser light scattering (SEC-MALLS) and thermogravimetric analysis (TGA). The M/G ratio of alginate was calculated by using proton nuclear magnetic resonance ($^1$H NMR). The antioxidant activity of mangosteen rind and the combination extracts was determined using the DPPH method. The observed parameters were wound width, number of neutrophils, macrophages, fibrocytes, fibroblasts, and collagen densities. The 36 male mice were divided into 12 groups including non-diabetic control (NC), diabetes alginate (DA), alginate–mangosteen (DAM), and diabetes control (DC) groups in three different groups by a histopathology test on skin tissue. The treatment was carried out for 14 days and mice were evaluated on Days 3, 7, and 14. The SEC-MALLS results showed that the molecular weight and dispersity index (Đ) of alginate were $2.77 \times 10^4$ Dalton and 1.73, respectively. The M/G ratio of alginate was 0.77 and described as single-stage decomposition based on TGA. Alginate, mangosteen rind extract, and their combination were divided into weak, medium, and strong antioxidant, respectively. The treatment of the DA and DAM groups showed a decrease in wound width and an increase in the number of fibrocytes, fibroblasts, and macrophages. The number of neutrophils decreased while the percentage of collagen densities increased for all the considered groups.

**Keywords:** alginate; *Sargassum ilicifolium*; mangosteen rind; wound healing; diabetic mice

## 1. Introduction

Diabetes mellitus (DM) is a metabolic disease caused by the disruption of the glucose metabolism in the body [1]. DM can cause changes in skin homeostasis resulting in

changes in metabolism and several complications, such as vasculopathy and neuropathy [2]. Diabetes is usually accompanied by the emergence of foot ulcer disease or gangrene, palpable femoral and popliteal pulses, and the absence of foot pulses [3]. Foot ulcer disease and gangrene are cases of disease that often occur in people with DM and usually begin with the appearance of wounds. In general, there are many ways to treat open wounds in diabetic patients: one of them being the use of ointments as topical medication. Agrawal et al. (2014) reported that wounds can be healed using antibiotics, either used topically, directly on the wound, or orally [4]. Diabetic wound healing can be disrupted if the patient has poor blood sugar control and therapy, as well as the presence of bacteria on the wound surface that causes infection [5]. In fact, the problems that occur during the wound-healing process make the wound worsen, e.g., by increasing the reactive oxygen species (ROS) level and oxidative stress. Both of these problems can disrupt the wound-healing process, so the healing takes longer. In normal conditions, ROS such as hydrogen peroxide ($H_2O_2$) and superoxide act as cellular messengers to stimulate wound healing [6]. Increasing the level of ROS has a beneficial effect but, in some cases, can also cause tissue damage [7]. When diabetic complications occur, the ROS level and oxidative stress start causing cell death and tissue damage via several mechanisms [8]. Disproportion between antioxidants and ROS disturbed by a depletion of antioxidants or an accumulation of ROS causes oxidative stress [9]. This situation might be resolved through the application of antioxidants. Lobo et al. (2010) reported that antioxidant molecules contribute an electron to make a free radical become neutral, thus reducing its capacity for cellular damage through their free radical scavenging property [10].

Alginate is a natural polymer that contains β-D-mannuronic acid (M) and α-L-guluronic acid (G) blocks [11,12]. The main source of alginate is the various genera of brown seaweed [12]. Alginate as a topical medicinal material has been chosen because it is considered capable in maintaining the humidity around the wound, minimizing bacterial infections, and easing the wound-healing process [13]. Alginate has been used in several types of wounds, including pressure, diabetic, venous ulcers, and some open wounds due to its ability as a good absorbent and gel-forming agent [14,15].

Mangosteen rind has been a traditional medicine to treat skin trauma and infections for many decades [16]. Several studies have reported that xanthones (such as α-mangostin) are the main compounds found in mangosteen rind. Xanthone has antibacterial, anti-inflammatory, antioxidant, anticancer, and cardioprotective activities [17]. In the present paper, the histopathology of wound healing on a diabetic mice model by using a composite of alginate from *S. duplicatum* and okra combination extracts is described. Here, we report the wound-healing and antioxidant evaluations of alginate from *S. ilicifolium* and mangosteen rind combination extracts on a diabetic mice model.

## 2. Materials and Methods

### 2.1. Materials

*S. ilicifolium* samples were taken from Kei Island, Maluku, Indonesia in July 2018. Identification was performed at the Oceanographic Research Center, LIPI, Jakarta, Indonesia (Approval Reference Number: 1/3/18-id/2018). Mangosteen (*Garcinia mangostana*) was purchased from the Rungkut district (Surabaya, Indonesia). The species identification was carried out at the Biosystem Laboratory of the Department of Biology of Airlangga University.

### 2.2. Sodium Alginate Extraction

The *S. ilicifolium* was dried, cut into pieces, and grounded to powder. The dried *S. ilicifolium* was added in a Becker in 0.1% KOH, soaked for an hour, and filtered. Then, 1% HCl (1:30 *w/v*) was added to the sample and stirred for an hour, then it was filtered and washed using water until a neutral pH was reached. The sample was then soaked with 2% $Na_2CO_3$ at $\pm70\,^\circ$C for two hours and filtered. The filtrate was added with 10% HCl to pH 2.8–3.2 or until a gel was formed and 2% $Na_2CO_3$ until neutral pH was added to

the gel. Then, 4% NaOCl (1:2 *v/v*) was added to the mixture. The alginate extract was pipetted into 2-propanol solution in a 1:2 *v/v* ratio and slowly stirred until a sodium alginate fiber was formed. The sodium alginate fiber was dried and grounded to form sodium alginate powder.

### 2.3. Analysis of Sodium Alginate

2.3.1. Proton Nuclear Magnetic Resonance ($^1$H NMR) Spectroscopy

$^1$H NMR spectroscopy analyses were conducted on a JEOL JNM-ECS400A spectrometer (JEOL, Peabody, MA, USA) at a frequency of 400 MHz. D$_2$O was used as the NMR solvent if not otherwise detailed. All samples were freeze dried before analysis.

2.3.2. Size Exclusion Chromatography with Multi-Angle Laser Light Scattering (SEC-MALLS) Analysis

The molecular weight of alginate and its classification were analyzed using the gel permeation chromatography technique by using a Shimadzu HPLC system (Shimadzu UK Limited, Milton Keynes, UK) comprising a CBM-20A Controller, LC-20AD Pump with degasser, SIL-20A Autosampler, and SPD-20A detector; HELEOS-II light-scattering and Optilab rEx refractive index detectors were provided by Wyatt. A PL aquagel-OH mixed column (7.5 × 300 mm, 8 μm particle size; Agilent Technologies, Santa Clara, CA, USA) and a PL aquagel-OH mixed guard column (7.5 × 50 mm, 8 μm particle size; Agilent Technologies, Santa Clara, CA, USA) were used as the stationary phase. The mobile phase was a 50 mM sodium nitrate solution with a flow rate of 0.5 mL/min [11]. The alginate solution (1.5 mg mL$^{-1}$), dissolved in deionized water, was filtered through a nylon membrane (Whatman, UK) before analysis. The value of 0.165 was used as the dn dc$^{-1}$ grade [18].

2.3.3. Thermogravimetric Analysis (TGA)

TGA was carried out on a PL Thermal Sciences STA 625 thermal analyzer (PL Thermal Science Limited, Surrey, UK). A total of 10 mg of accurately weighed sample in an aluminum sample cup was placed into the furnace with a N$_2$ flow of 100 mL min$^{-1}$ and heated from room temperature to 625 °C at a heating rate of 10 °C min$^{-1}$. From the TGA profiles, the temperatures at 10% and 50% mass loss (TD$_{10}$ and TD$_{50}$, respectively) were analyzed afterwards.

### 2.4. Extraction of Mangosteen Rind

The dry mangosteen rinds were mashed into small fragments. The fragments were then macerated using 96% ethanol for three days. Samples were filtered and concentrated under vacuum. The extract was freeze dried and kept in 4 °C for the next step.

### 2.5. Antioxidant Assay and Analysis of Total Phenolic Content

Antioxidant assays were performed on alginate from *S. ilicifolium*, mangosteen rind extract, and a combination of alginate–mangosteen rind extracts by the 2,2-diphenyl-1-picryl-hydrazyl-hydrate (DPPH) free radical method. The concentration of DPPH and stock solutions were 50.0 and 1000.0 μg/mL, respectively. The extracts were diluted in concentration variations of 200.0, 150.0, 125.0, 100.0, 75.0, 50.0, 35.0, 25.0, 15.0, 10.0, and 6.0 μg/mL of sample solution in methanol (200 μL) and 50 μL of DPPH was added into 96-well plates. Subsequently, the mixtures were incubated for 30 min in a dark chamber and the absorbance was measured at 517 nm on an ELISA plate reader.

The total phenolic content of alginate and mangosteen rind extract was determined by the colorimetric method, referring to the procedure of Malik et al. (2015) with some modifications and with gallic acid as standard [19]. The standard solutions and samples with concentrations of 10, 20, 30, 40, and 50 ppm were added with 0.4 mL of Folin–Ciocalteau reagent. After 4–8 min, 4.0 mL of 7% Na$_2$CO$_3$ solution was added and aquabidestilata was added to 10 mL, and then it was allowed to stand for 2 h at room temperature. The

absorbance of solution was analyzed at a maximum wavelength of 744.8 nm; a calibration curve was made for the relationship between the concentration of gallic acid (μg/mL) and the absorbance.

### 2.6. Induction of Diabetic Mice

The BALB/c strains of adult male mice (*Mus musculus*) were obtained from the Faculty of Pharmacy, Airlangga University, Surabaya. The mice were 3–4 months old with a weight range of 20–35 g. Ethical clearance for treating animals was obtained from the Faculty of Veterinary Medicine, Airlangga University, Surabaya, Indonesia with License Reference Number: 2. KE. 049.04.2019.

The mice were treated in a 12 h light and 12 h dark lighting system for a couple of weeks. During acclimatization for the experiment, the mice were fed (oral dose of 0.3 mL) with lard three times per week during 3 weeks in order to achieve a high fat diet. Streptozotocin (STZ) was injected for 8 days by using the 30 mg/kg body weight using multiple low dose method of 0.15 mL intraperitoneally (i.p) to induce type II diabetes mellitus [20]. The mice were weighed before and after treatment. The level of blood glucose was measured after STZ induction at 3, 7, and 14 days of treatment.

### 2.7. Animal Grouping and Treatment

The group of mice (12 groups) were separated: (a) three groups for no treatment as normal (N) and (b) nine diabetic groups for treatment (D), consisting of three mice in each group for 3, 7, and 14 days of treatment [20]. The nine treatment groups were classified into three categories: (1) three groups for diabetic control (DC3, DC7, and DC14); (2) three groups for alginate treatment (DA3, DA7, and DA14); and (3) three groups for alginate–mangosteen rind extract treatment (DAM3, DAM7, and DAM14). A centimeter wound on the mice's *glutea* (buttocks) was generated. Each group was smeared with Vaseline (untreated and DC), vaseline–alginate (DA), and vaseline–alginate–mangosteen rind extract ointments (DAM). A single dose of 50 mg/kg body weight ointment was generated for treatment. The animal number replication was referred to the Federer formula (1967) [21].

### 2.8. Histopathology Analysis of Wound Healing

The histopathology experiments of wound healing were conducted at the Pathology Laboratory of the Faculty of Veterinary Medicine, Airlangga University. A microscope at 400 times magnification was used for the observation of fibroblasts, fibrocytes, macrophages, neutrophils, and collagen density. The width of the wound was analyzed using 40 times magnification under an optical microscope.

### 2.9. Statistical Analysis

The wound-healing measurements are illustrated as a mean ± standard error mean (SEM). The wound-healing data analysis was conducted by a normality and homogeneity test, one-way ANOVA, and Duncan test. If the results of normality and homogeneity were not qualified ($\alpha = 0.05$), non-parametric tests such as the Kruskal–Wallis and Mann–Whitney tests were applied. All the statistical analyses were calculated using an IBM with SPSS 20.0 software.

## 3. Results

### 3.1. Analysis of Sodium Alginate

#### 3.1.1. Proton Nuclear Magnetic Resonance ($^1$H NMR) Analysis

The mannuronate-to-guluronate (M/G) ratio of alginate was determined by $^1$H NMR. The spectrum of alginate is shown in Figure 1. The proton signals of alginate at a chemical shift in the 4.8–4.0 ppm range correspond to the $G_2/M_2$, while the signals at 5.8–5.0 ppm correspond to the $G_1/M_1$, respectively. The $G_1$ and $M_1$ are the most de-shielded protons ($\delta$ 5.8–5.1 and 5.5–5.1, respectively). $M_1$ is more shielded than $G_1$, because $G_1$ is located at

an equatorial position in comparison to an axial one in $M_1$. The M/G ratio of alginate was 0.77 and was determined by the integration as previously reported [22].

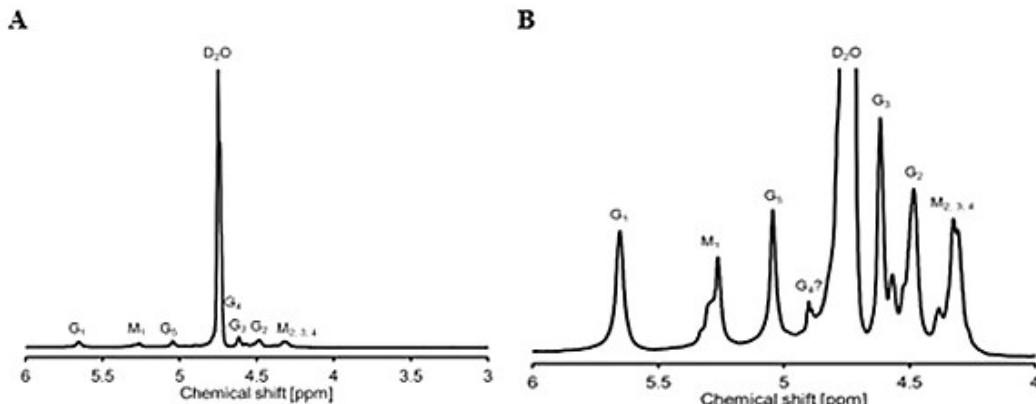

**Figure 1.** (**A**) Spectra of $^1$H NMR spectra of alginate from *S. Ilicifolium* (90 °C 128 scans); (**B**) detail of $^1$H NMR spectra (90 °C 128 sc).

### 3.1.2. Size Exclusion Chromatography with Multi-Angle Laser Light Scattering (SEC-MALLS) Analysis

The polydispersity index (Đ) and the molecular weight of alginate from *S. ilicifolium* were analyzed by SEC-MALLS (Figure 2A). The number average molecular weight ($M_N$) and weight average molecular weight ($M_W$) of alginate from *S. ilicifolium* were $1.49 \times 10^4$ and $2.77 \times 10^4$ Dalton, respectively. The Đ was found to be 1.73, which indicated that alginate from *S. ilicifolium* has a good homogeneity and is classified as chain-growth polymerization.

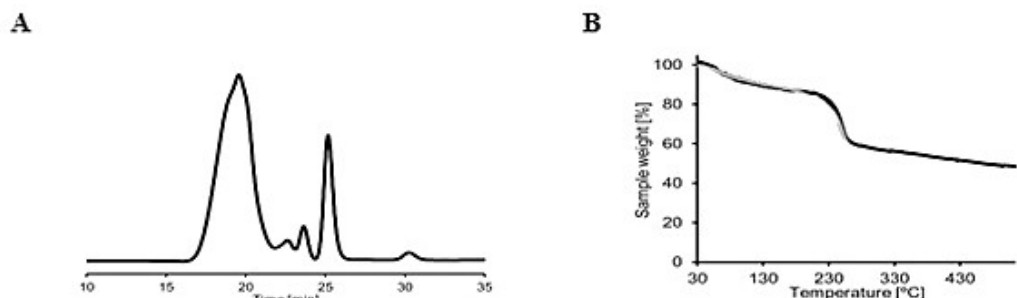

**Figure 2.** (**A**) RI signal of size exclusion chromatography with multiple angle laser light scattering (SEC-MALLS) analysis and (**B**) thermogravimetric analysis (TGA) of sodium alginate.

### 3.1.3. Thermogravimetric Analysis

Thermal stability and polymer matrix strength were analyzed using TGA by observing the weight difference as the sample was heated at a constant rate. Figure 2B shows the thermal stability and strength of the *S. ilicifolium* alginate polymer matrix. It was found that 5% of the mass was obscured ($TD_5$) at 82 °C and most probably indicated the water loss (moisture). The 10% ($TD_{10}$) and 20% mass loss occurred at 125 °C and 235 °C, respectively, indicating glycosidic bond destruction. The 50% ($TD_{50}$) mass loss at 466 °C indicated the carbon turns into charcoal.

### 3.2. Antioxidant Assay and Analysis of Total Phenolic Content

An antioxidant assay was utilized to measure the inhibitory concentration ($IC_{50}$) of each compound. The antioxidant activity of alginate, mangosteen rind, and the combination of alginate–mangosteen rind extracts was found using the DPPH method. The $IC_{50}$ of

alginate, mangosteen rind, and the combination of alginate–mangosteen rind extract was 297.60, 29.60, and 52.72 μg/mL, respectively (Table 1). The results showed that alginate was a moderate antioxidant, mangosteen rind was strong antioxidant, and the combination of alginate–mangosteen rind extracts was acting as a very powerful antioxidant. The total phenolic content in the mangosteen rind extract was 32.94 mg GAE/g extract or 3.29% (y = 0.0193x + 0.031; $R^2$ = 0.9976).

**Table 1.** Antioxidant activity of alginate, mangosteen rind, and alginate–mangosteen rind combination.

| Sample | $IC_{50}$ (μg/mL) | Classification |
|---|---|---|
| *S. ilicifolium* alginate | 120.88 | Moderate antioxidant |
| Mangosteen rind extract | 29.06 | Powerful antioxidant |
| Alginate–mangosteen rind extract | 52.72 | Strong antioxidant |

### 3.3. Measurement of Body Weight in Mice

The administration of lard for three weeks in a single oral dose (0.3 mL) can significantly increase the body weight of mice ($p < 0.05$). The increase in the body weight of mice ranged from 29 ± 3.42 g to 34 ± 2.83 g. Husen et al. (2019) reported that obesity caused by excessive fat accumulation can induce various chronic diseases and complications such as diabetes mellitus [23].

### 3.4. Measurement of Blood Glucose Level in Mice

STZ can increase blood sugar levels of up to more than 250 mg/dL in mice, as presented in Table 2. Based on experiments, the blood sugar levels of the normal group decreased significantly on Days 1 to 14 ($p < 0.05$). The blood glucose level in the DC group decreased from Days 1 to 7 but increased on Day 14 (250.0 ± 10.6 mg/dL). The DA group showed a decrease from Days 1 to 14 (250.0 ± 8.5 mg/dL to 209.0 ± 11.3 mg/dL). Fluctuating changes in body weight were observed in the DAM mice group, but on Day 14 the blood sugar level decreased to 135.0 ± 7.1 mg/dL. On Day 14, the DAM mice showed a normal blood sugar level (non-diabetic).

**Table 2.** Mean and standard deviation of blood glucose.

| Group Treatment | Blood Glucose Level (mg/dL) | | | |
|---|---|---|---|---|
| | Day 1 | Day 3 | Day 7 | Day 14 |
| Normal (NC) | 118.0 ± 4.2 | 117.0 ± 2.3 | 114.0 ± 2.8 | 110.0 ± 1.4 |
| Diabetic (DC) | 256.0 ± 4.9 | 250.0 ± 11.3 | 242.0 ± 2.1 | 250.0 ± 10.6 |
| Alginate ointment (DA) | 250.0 ± 8.5 | 230.0 ± 12.0 | 225.0 ± 2.8 | 209.0 ± 11.3 |
| Alginate–mangosteen rind extract ointment (DAM) | 251.0 ± 5.7 | 176.0 ± 5.7 | 182.0 ± 7.8 | 135.0 ± 7.1 ** |

Asterisk notation in the table shows the result of the statistical test ($\alpha < 0.05$). ** notation shows insignificant differences with NC but significant differences with DC. ** = $0.01 \leq \alpha < 0.05$. $n$ = 3 animals per time point.

### 3.5. Microscopic Evaluation of Wound Tissue

The microscopic evaluation of wound tissue included an evaluation of wound width, the number of neutrophil cells, macrophages, fibrocytes, and fibroblasts, and the collagen density in each group. The results of the histopathological microscopic observations in all study groups are described in Figures 3 and 4.

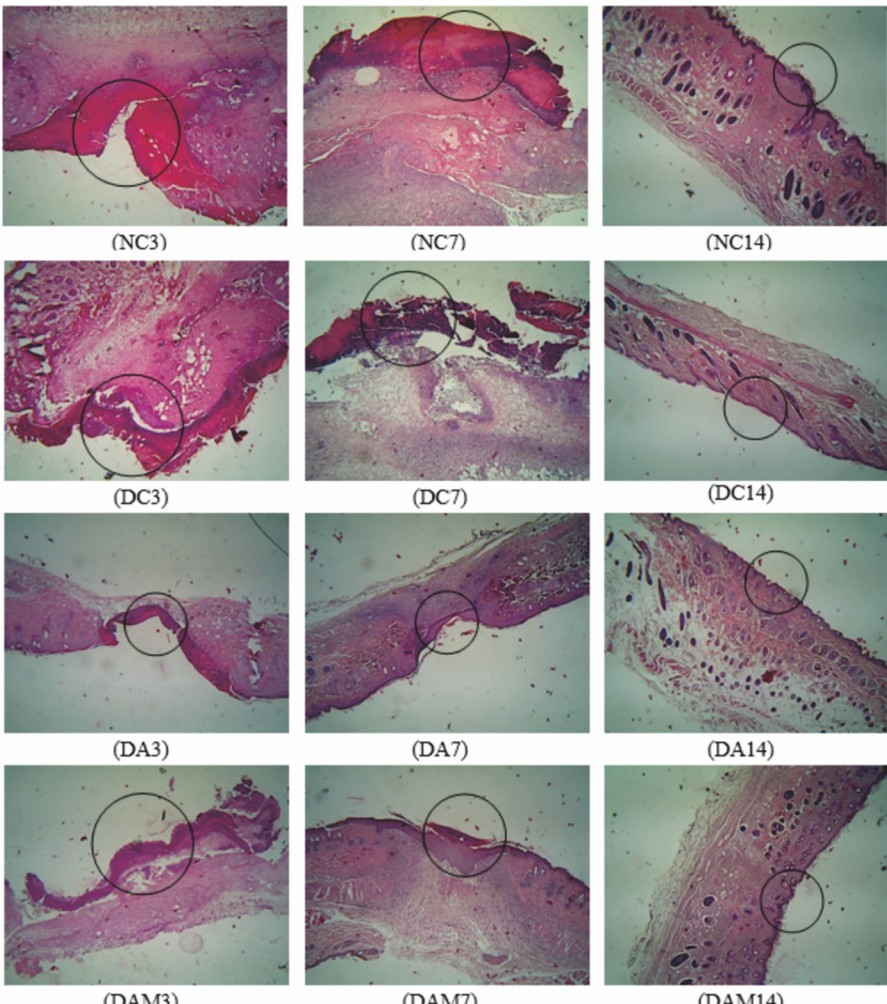

**Figure 3.** Histopathology analysis of wound width was measured on the three variation times. A circle in the picture indicates the wound width in the specimen. NC: normal group; DA: alginate treatment group; DAM: alginate–mangosteen rind extract treatment group; and DC: diabetic group.

Based on microscopic observations, it can be seen that the DAM group revealed the best wound-healing process for diabetic mice compared to other diabetic treatment groups. The re-epithelization of DA and DAM occurred on Day 7 and wound closing occurred on Day 14 for all groups. Figure 4 shows that the DAM group had the best re-epithelization process among the groups.

Figure 4 shows the wound parameters in microscopic observation, where neutrophils (green), macrophages (blue), fibrocytes (yellow), and fibroblasts (red) are shown by an arrow in a different color. The number of neutrophils increased at Day 3 and decreased at Day 7 and Day 14 for all groups. The number of macrophages increased at Day 3 to Day 7, and then declined at Day 14 for all groups. NC, DA, and DAM showed a significant rise in the number of fibrocytes and fibroblasts at Day 7 to indicate re-epithelization and a reduction at Day 14.

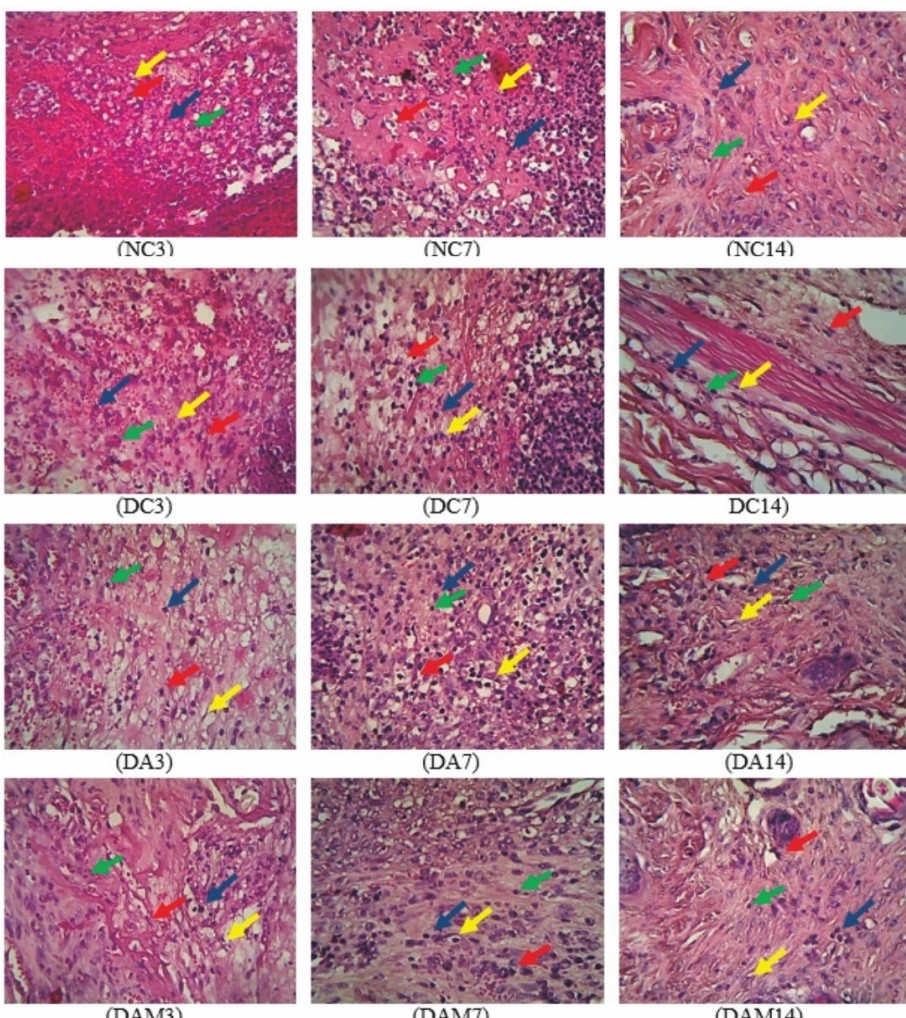

**Figure 4.** Analysis of wound width on the three variation times. Green arrows: neutrophils; blue arrows: macrophages; yellow arrows: fibrocytes; and red arrows: fibroblasts. NC: normal group; DA: alginate treatment group; DAM: alginate–mangosteen rind extract treatment group; and DC: diabetic group.

### 3.6. Parameter Analysis

### 3.6.1. Wound Width Determination

The width of the wound is defined as the distance between the epithelium (right–left) that underwent a complete re-epithelialization quantifiable in microns. The width of the wound for each group per day showed a significant difference (Figure 3). All groups showed good results on Day 14 as the wound had closed in all groups. DAM showed the best wound repair response in comparison with the NC group (2502.0 ± 10.4 μm and 726.0 ± 29.7 μm on Days 3 and 7, respectively) (Figure 5).

### 3.6.2. Neutrophil Measurement

Neutrophils are the onset of an inflammatory response in the recovery process of open wounds and aim to clean microorganisms that enter the wound [24–26]. The number of neutrophils showed significant differences both in the DA and DAM groups toward DC. A decrease in the neutrophil count was observed in DAM from Days 3 to 14, similar to what was observed for the NC group (Figure 6).

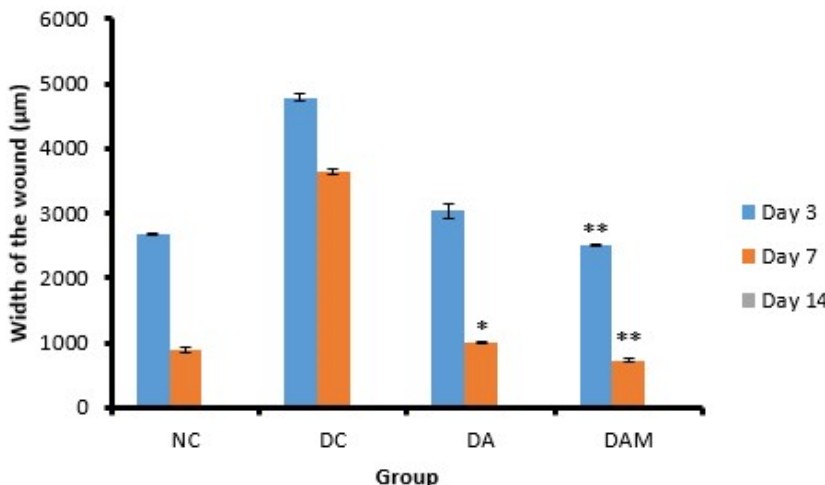

**Figure 5.** Wound width at the three variation times. Notation on the graph shows the result of the statistical test ($\alpha < 0.05$). * and ** notation shows insignificant differences with NC but significant differences with DC. * = $0.01 \leq \alpha < 0.05$; ** = $\alpha \leq 0.01$. NC: normal group; DA: alginate treatment group; DAM: alginate–mangosteen rind extract treatment group; and DC: diabetic group.

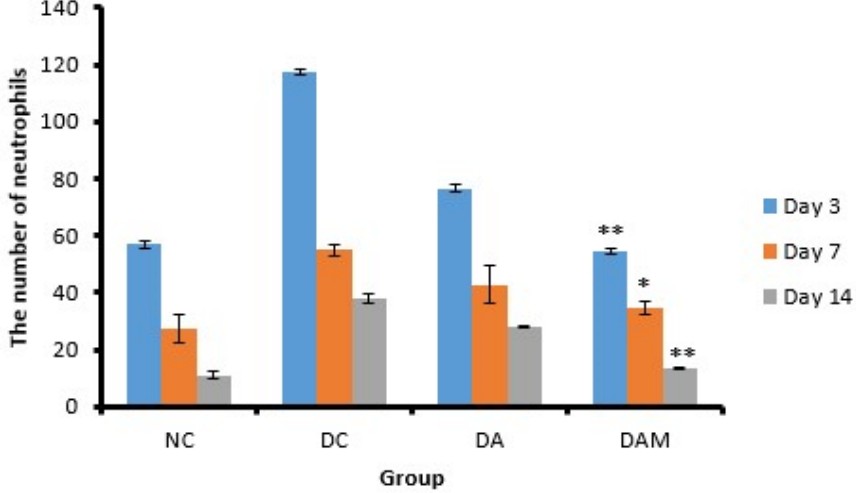

**Figure 6.** Number of neutrophils at the three variation times. Notation on the graph shows the result of the statistical test ($\alpha < 0.05$). * and ** notation shows insignificant differences with NC but significant differences with DC. * = $0.01 \leq \alpha < 0.05$; ** = $\alpha \leq 0.01$. NC: normal group; DA: alginate treatment group; DAM: alginate–mangosteen rind extract treatment group; and DC: diabetic group.

### 3.6.3. Macrophage Measurement

Macrophage secretion will increase massively as neutrophils decrease [26]. Figure 7 shows that the number of macrophages in the DC group showed a high increase due to hyperglycemia and ROS, while the other groups experienced a decrease because they did not experience these conditions. The DAM group gave a similar observation as the NC group, wherein the graph of macrophages in this group was the same as the control normal/non-diabetic group (Figure 7).

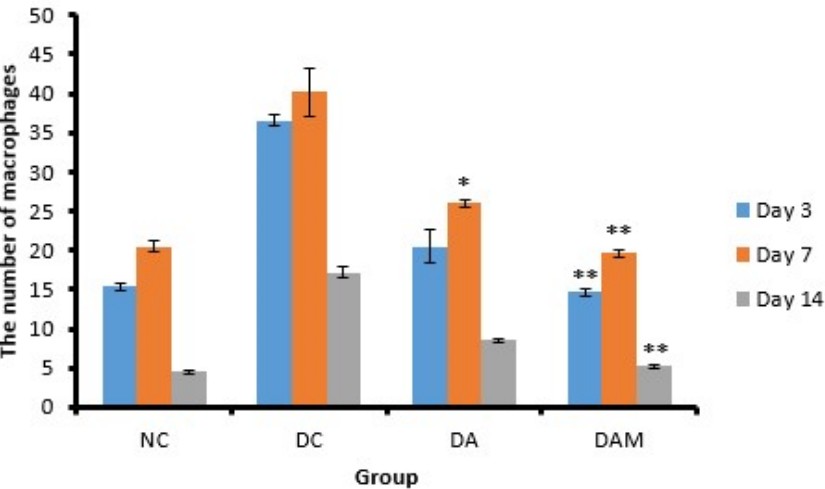

**Figure 7.** Number of macrophages at the three variation times. Notation on the graph shows the result of the statistical test ($\alpha < 0.05$). * and ** notation shows insignificant differences with NC but significant differences with DC. * = $0.01 \leq \alpha < 0.05$; ** = $\alpha \leq 0.01$. NC: normal group; DA: alginate treatment group; DAM: alginate–mangosteen rind extract treatment group; and DC: diabetic group.

### 3.6.4. Fibrocyte Measurement

Fibrocytes play a role in the process of differentiation of myofibroblasts in fibrosis and wound healing [27]. In this study, the number of fibrocytes in the DAM groups on Days 3, 7, and 14 were close to the NC group, whereas the DC group showed the lowest number of fibrocytes, i.e., almost one-third of the numbers recorded for the NC and DAM groups (Figure 8).

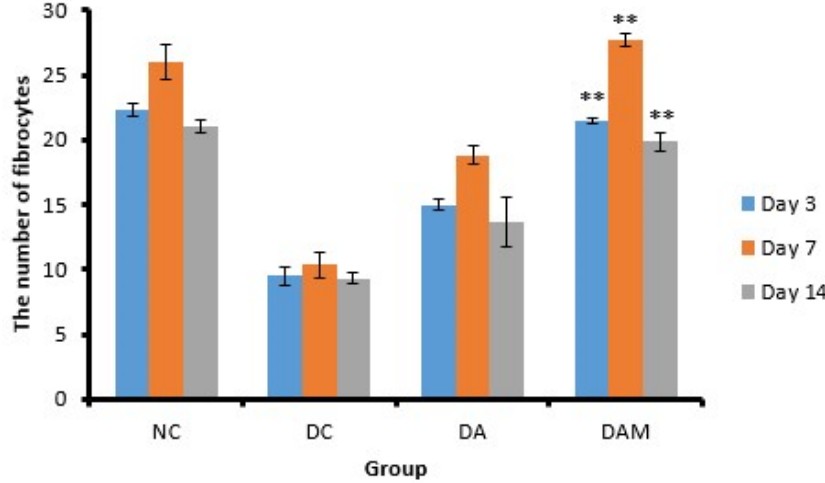

**Figure 8.** Number of fibrocytes at the three variation times. Notation on the graph shows the result of the statistical test ($\alpha < 0.05$).** notation shows insignificant differences with NC but significant differences with DC. ** = $\alpha \leq 0.01$. NC: normal group; DA: alginate treatment group; DAM: alginate–mangosteen rind extract treatment group; and DC: diabetic group.

### 3.6.5. Fibroblast Measurement

The histopathological observations are displayed in Figure 4. DAM showed the highest number of fibroblasts, while DC showed the lowest number of fibroblasts (Figure 9). The number of fibroblasts on Days 3 and 7 showed an increase but decreased on Day 14 in all groups (Figure 9).

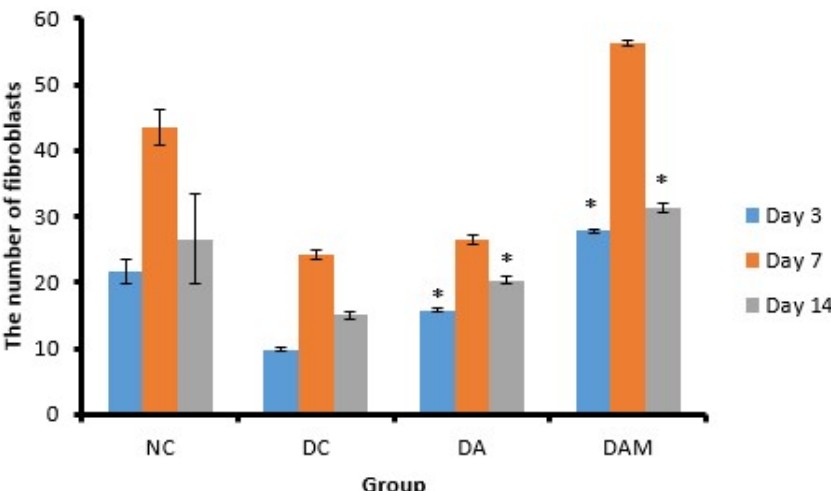

**Figure 9.** Number of fibroblasts at the three variation times. Notation on the graph shows the result of the statistical test ($\alpha < 0.05$). * notation shows insignificant differences with NC but significant differences with DC. * = $0.01 \leq \alpha < 0.05$. NC: normal group; DA: alginate treatment group; DAM: alginate–mangosteen rind extract treatment group; and DC: diabetic group.

3.6.6. Collagen Density Measurement

Observation of collagen density was conducted to determine the process of the formation of new collagen in open wounds. Figure 10 shows that DAM presents the highest collagen density compared to the other groups, while DC shows the lowest percentage of collagen density among the considered groups.

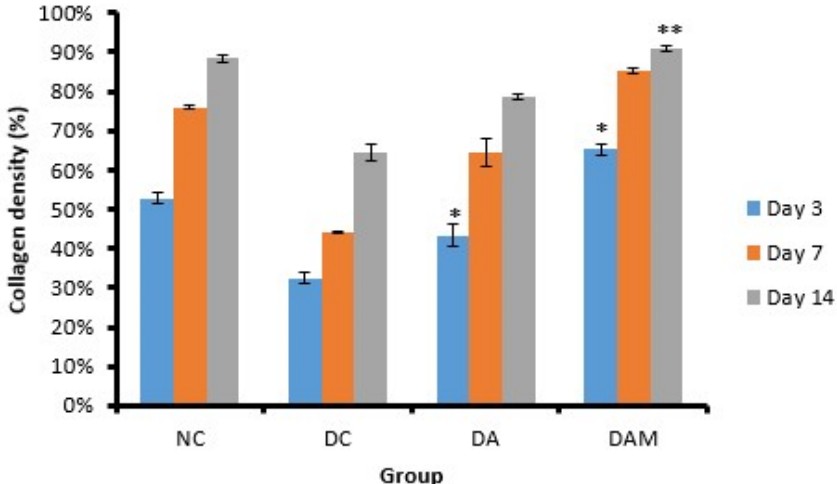

**Figure 10.** Collagen density was evaluated at the three variation times. Notation on the graph shows the result of the statistical test ($\alpha < 0.05$). * and ** notation shows insignificant differences with NC but significant differences with DC. * = $0.01 \leq \alpha < 0.05$; ** = $\alpha \leq 0.01$. NC: normal group; DA: alginate treatment group; DAM: alginate–mangosteen rind extract treatment group; and DC: diabetic group.

## 4. Discussion

Sodium alginate was extracted from *S. ilicifolium* to give a pale-yellow solid with a 14.36% yield. The obtained yield is very similar to the one reported by Latifi et al. (2015), who reported yields ranging from 12% to 16.5% [28]. Davis et al. (2004) extracted *S. fluitans* and *S. oligocystum* using high temperature and base conditions to obtain 21.1–22.8% and 18.9–20.5% yields, respectively [29]. Mushollaeni (2011) reported that alginate from several species of *Sargassum* sp. had 16.93 to 30.50% yields [30]. Based on this study, the yield of the

extracted alginate not only depends on the extraction method and the *Sargassum* species used but also on the seasonal variation. It is in fact well known that there are marked seasonal variations of polysaccharide contents in seaweeds, as polysaccharides are stored in winter as a food source and therefore a harvest at the end of summer will give higher polysaccharide levels than in the spring.

The $^1$H NMR analysis of *S. ilicifolium* alginate showed that the signals in the range 4–4.8 ppm belong to protons from the β-glycosidic bond, whereas the signals between 5.1 and 5.8 ppm are resonances of protons from the α-glycosidic bond. The identification was performed by comparison with the $^1$H NMR spectrum of alginate from Llanes et al. (1997) [31]. The structure of alginate from *S. ilicifolium* consists of β-D-mannuronic acid (4.3–4.8 ppm) and α-L-guluronic acid (5.1–5.8 ppm). The M/G ratio of alginate was determined by the integration of each proton with their monomers. The M/G ratio of alginate was 0.77, indicating that *S. ilicifolium* alginate has a higher guluronic acid monomer than mannuronic acid. The protons $G_1$ and $M_1$ are the most de-shielded protons (δ 5.1–5.8 and 5.1–5.5, respectively). $M_1$ is more shielded than $G_1$, because $G_1$ is located at an equatorial position in comparison to an axial one in $M_1$. In fact, Llanes et al. performed a pretreatment to partly hydrolyze the alginate before $^1$H NMR analysis, while in our case we decided to work with non-hydrolyzed alginate as it was fully soluble in the NMR solvent. This can be an explanation for the slightly more structured peaks and possibly also for the small shifts downfield we observed.

The number of $M_n$ and $M_w$ of alginate were $1.49 \times 10^4$ and $2.77 \times 10^4$ Dalton, respectively. Furthermore, the $M_n$ and $M_w$ obtained were used in the calculation of the dispersity index (Đ) using the following equation:

$$Đ = Mw/Mn \tag{1}$$

The dispersity index (Đ) was 1.73. The Đ value of alginate from *S. ilicifolium* was below 2, indicating that the extracted alginate showed good homogeneity and was classified as chain-growth polymerization. Chain-growth polymerization is a process in which the high molecular weight polymer is generated at the beginning of the polymerization process and the polymer yield (the percentage conversion of the monomer into the polymer) gradually increases with time [32]. Based on TGA, alginate from *S. ilicifolium* had a single-stage decomposition curve (Figure 2B).

The antioxidant assay showed that the $IC_{50}$ of alginate, mangosteen rind, and alginate–mangosteen rind extract combination was 297.60, 29.60, and 52.0 μg/mL, respectively. The results indicated that alginate was a weak antioxidant and mangosteen rind was a powerful antioxidant, and when combined they produced a material with excellent antioxidant properties. These observations could be linked to the fact that mangosteen rind contains xanthones, such as α-mangostin and γ-mangostin, which have been reported to exhibit anti-inflammatory, antioxidant, and radical scavenging activity [33–35]. According to Thong et al. (2015), xanthones can capture free radicals in two ways: through hydrogen atom transfer (HAT) to explain the antioxidant activity in the gas phase and the single electron transfer–proton transfer (SETPT) mechanism that is thermodynamically favored in water [36]. Strong antioxidant activity observed in the combination of alginate and mangosteen rind extract could possibly overcome ROS that occur in hyperglycemic conditions, thus enhancing wound repair in diabetic mice.

The total phenolic content in mangosteen rind extract tends to be low due to several factors, such as the type of mangosteen fruit, storage time, and the hardness of the rind. According to Dangcham et al. (2008), the total phenolic content of mangosteen fruit will decrease throughout the storage time [37]. Another study reported that the lignin content and density of the damaged pericarp tissue would increase as a result of the hardening of the fruit skin, while the total phenolic content decreased [38]. Mangosteen rind is an essential, natural phenolic antioxidant source; at least ten phenolic acids have been characterized in mangosteen rind [39,40]. Mangosteen rind contains bioactive substances,

such as phenolic acids, flavonoids, anthocyanins, proanthocyanidins, (-)-epicatechin, and xanthones, which have biological, medical, and antioxidant properties.

The administration of lard can significantly increase the body weight of mice. This condition indicated increasing hyperglycemia and insulin resistance, and it will cause type II DM [20]. Husen et al. (2019) reported that obesity caused by excessive fat accumulation can induce various chronic diseases and complications such as diabetes mellitus [23]. The administration of STZ was performed to increase the condition of hyperglycemia (increased blood sugar level) [20]. The increase of blood sugar levels in three diabetic groups, namely diabetic non-treated (DC), diabetic treated with alginate ointment (DA), and diabetic treated in alginate–mangosteen rind extract ointment (DAM), of up to 250 mg/dL indicated that the mice were in a diabetic condition. Based on experiments, the combination of alginate and mangosteen rind extract proved able to reduce blood sugar levels (Table 2). Lee et al. (2018) reported that mangosteen was able to increase insulin secretion in pancreatic B-cells and protected cells from apoptosis, due to the presence of $\alpha$-mangostin in mangosteen fruit [41]. Jariyapongskul et al. (2015) reported that daily $\alpha$-mangostin supplementation in diabetic rats shows remarkable hypoglycemic and insulinotropic effects, a reduction in plasma glycated hemoglobin, and a decrease in serum triglycerides and cholesterol [42].

It was expected that the administration of a topical combination of alginate and mangosteen rind extract in diabetic wounds could reduce blood sugar levels by entering the systemic circulation through the bloodstream. The entry of compounds found in topical drugs through the bloodstream can enhance the host insulin level and the sensitivity of somatic cells [43]. Therefore, the topical combination of alginate and mangosteen rind extract can be classified as diadermic ointment. A diadermic ointment is a drug that is able to enter the deepest skin tissue and enter the systemic circulation [44]. However, further investigation is required to confirm whether the compounds responsible for reducing the blood sugar levels are a combination of alginate and mangosteen rind extract or the presence of other active compounds that have not been separated from the alginate during extraction given the number of weak proton signals observed in the [1]HNMR. In addition, it is necessary to confirm the exact mechanism concerning how blood glucose levels can be decreased by administering a topical combination of alginate and mangosteen rind extract.

The DAM group, which was treated by alginate–mangosteen rind extract combination, showed a very significant decrease on Days 3 and 7, $2502.0 \pm 10.4$ and $726.0 \pm 29.7$, respectively, compared to other groups (Table 3). On Day 14, all groups showed that the wound had closed. The wound width of diabetic mice in the DAM group showed the best activity when compared to the other tested conditions. This happened due to the combination of alginate and mangosteen rind extract providing strong antioxidant activity and thus decreasing ROS and increasing wound contraction and wound healing.

The diabetic group (DC) showed the greatest increase in the number of neutrophils in comparison to other groups. Administration of the combined alginate and mangosteen rind extracts (DAM) demonstrated that the neutrophil count in open wounds on diabetic mice increased in the first 24–48 h and then decreased until Day 14 ($55.0 \pm 0.9$ on Day 3 to $14.0 \pm 0.2$ cells/mm$^2$ on Day 14), thus indicating that DAM could ease the process of wound healing (Table 4). The decrease in the neutrophil count in the DAM group occurred due to the presence of phenolic compounds from the mangosteen rind extract that acted as antioxidants. Phenolic compounds are able to reduce the secretion of human neutrophil elastase (HNE), which is an enzyme that can break down the components of the extracellular matrix by reducing the number of neutrophils [45].

The macrophage count in the DC group showed a slight decrease on Day 14 compared to the other groups, thus implying that the inflammation phase in the open wounds of diabetic mice occurred for a longer duration than the normal group. Persistent inflammatory conditions inhibit wound healing [46]. The presence of hyperglycemia and oxidative stress (increased blood sugar levels and ROS) can affect the modulation and polarization of macrophages, which could inhibit the healing process [27,47]. Therefore, the administration

of alginate and mangosteen rind extracts to diabetic wounds could reduce the number of macrophages in the wound almost as well as non-diabetic wounds (Table 5).

**Table 3.** Counts considering the mean and standard deviation of wound width.

| Group Treatment | Wound Width (μm) | | |
|---|---|---|---|
| | Day 3 | Day 7 | Day 14 |
| Normal (NC) | 2673.0 ± 22.9 | 882.0 ± 44.1 | 0.0 ± 0.0 |
| Diabetic (DC) | 4786.0 ± 63.9 | 3636 ± 37.0 | 0.0 ± 0.0 |
| Alginate ointment (DA) | 3030.0 ± 126.6 | 1002.0 ± 11.5 * | 0.0 ± 0.0 |
| Alginate–mangosteen rind ointment (DAM) | 2502.0 ± 10.4 ** | 726.0 ± 29.7 ** | 0.0 ± 0.0 |

Asterisk notation in the table shows the result of the statistical test ($\alpha < 0.05$). * and ** notation shows insignificant differences with NC but significant differences with DC. * = $0.01 \leq \alpha < 0.05$; ** = $\alpha \leq 0.01$.

**Table 4.** Cell counts considering the mean and standard deviation of neutrophils.

| Group Treatment | Neutrophil (cells/mm$^2$) | | |
|---|---|---|---|
| | Day 3 | Day 7 | Day 14 |
| Normal (NC) | 57.0 ± 1.4 | 28.0 ± 4.9 | 11.0 ± 1.4 |
| Diabetic (DC) | 118.0 ± 1.2 | 55.0 ± 1.9 | 38.0 ± 1.4 |
| Alginate ointment (DA) | 77.0 ± 1.2 | 43.0 ± 6.4 | 28.0 ± 0.4 |
| Alginate–mangosteen rind ointment (DAM) | 55.0 ± 0.9 ** | 35.0 ± 2.1 * | 14.0 ± 0.2 ** |

Asterisk notation in the table shows the result of the statistical test ($\alpha < 0.05$). * and ** notation shows insignificant differences with NC but significant differences with DC. * = $0.01 \leq \alpha < 0.05$; ** = $\alpha \leq 0.01$.

**Table 5.** Cell counts considering the mean and standard deviation of macrophages.

| Group Treatment | Macrophage (cells/mm$^2$) | | |
|---|---|---|---|
| | Day 3 | Day 7 | Day 14 |
| Normal (NC) | 15.0 ± 0.5 | 21.0 ± 0.7 | 5.0 ± 0.2 |
| Diabetic (DC) | 37.0 ± 0.7 | 40.0 ± 3.1 | 17.0 ± 0.7 |
| Alginate ointment (DA) | 21.0 ± 2.1 | 26.0 ± 0.5 * | 9.0 ± 0.2 |
| Alginate–mangosteen rind ointment (DAM) | 15.0 ± 0.5 ** | 20.0 ± 0.5 ** | 5.0 ± 0.2 ** |

Asterisk notation in the table shows the result of the statistical test ($\alpha < 0.05$). * and ** notation shows insignificant differences with NC but significant differences with DC. * = $0.01 \leq \alpha < 0.05$; ** = $\alpha \leq 0.01$.

An increase in the fibrocyte count leads to an increase in collagen V production, while lowering collagen I, III, and IV levels [48]. Fibrocyte differentiation can reduce inflammatory conditions and tissue damage as well as improve the wound-healing and tissue-remodeling process. The number of fibrocytes will increase with the increasing age of the wound, and more than 15 fibrocytes indicated the age of the wound was between 9 and 14 days [27]. The administration of a topical combination of alginate and mangosteen rind extract in the DAM group can increase the number of fibrocyte cells in diabetic open wounds between Day 3 and Day 7, thus accelerating the process of angiogenesis. On the other hand, the fibrocyte count decreased by Day 14, as shown in Table 6; this condition indicated the wound had healed.

**Table 6.** Cell counts considering the mean and standard deviation of fibrocytes.

| Group Treatment | Fibrocyte (cells/mm$^2$) | | |
|---|---|---|---|
| | Day 3 | Day 7 | Day 14 |
| Normal (NC) | 22.0 ± 0.5 | 26.0 ± 0.4 | 21.0 ± 0.5 |
| Diabetic (DC) | 10.0 ± 0.7 | 10.0 ± 0.9 | 9.0 ± 0.5 |
| Alginate ointment (DA) | 15.0 ± 0.5 | 19 ± 0.7 | 14.0 ± 1.9 |
| Alginate–mangosteen rind ointment (DAM) | 22.0 ± 0.2 ** | 28 ± 0.5 ** | 20.0 ± 0.7 ** |

Asterisk notation in the table shows the result of the statistical test ($\alpha < 0.05$). ** notation shows insignificant differences with NC, but significant differences with DC. ** = $\alpha \leq 0.01$.

Fibroblasts are cells that have a role in the proliferation phase. These cells have the function to regulate collagen, glycosaminoglycan, proteoglycan, fibronectin, and elastin as the extracellular matrix components [49,50]. In addition, fibroblasts also play a role in collagen. The presence of an increase and decrease in the number of fibroblasts is the same as observed in the measurement of the number of fibrocytes in the wound. All groups showed an increase in the number of fibroblasts on Day 7 and a decline on Day 14. A decrease in the number of fibroblasts on Day 14 in all groups indicates that the healing phase is at the remodeling stage (Table 7). The remodeling phase can be identified by decreasing proliferation and inflammation, reorganization of the extracellular matrix, and regression of newly formed capillary vessels [51]. The fibroblast count in the DAM group was almost the same as the fibroblast count of the NC or normal group. This result indicated that the administration of alginate–mangosteen rind extract combination was effective to heal the open wound.

**Table 7.** Cell counts considering the mean and standard deviation of fibroblasts.

| Group Treatment | Fibroblast (cells/mm$^2$) | | |
|---|---|---|---|
| | Day 3 | Day 7 | Day 14 |
| Normal (NC) | $22.0 \pm 1.9$ | $44.0 \pm 2.6$ | $27.0 \pm 6.8$ |
| Diabetic (DC) | $10.0 \pm 0.2$ | $24.0 \pm 0.7$ | $15.0 \pm 0.5$ |
| Alginate ointment (DA) | $16.0 \pm 0.2$ * | $27.0 \pm 0.7$ | $20.0 \pm 0.5$ * |
| Alginate–mangosteen rind ointment (DAM) | $28.0 \pm 0.2$ * | $56.0 \pm 0.5$ | $31.0 \pm 0.7$ * |

Asterik notation in the table shows the result of the statistical test ($\alpha < 0.05$). * notation shows insignificant differences with NC but significant differences with DC. * = $0.01 \leq \alpha < 0.05$.

The treatment group of alginate and mangosteen rind extract of DAM showed significant results at each observation on Days 3, 7, and 14 with $65.0 \pm 1.4\%$, $85.0 \pm 0.7\%$, and $91.0 \pm 0.7\%$, respectively (Table 8). The DAM group showed the highest percentage of collagen density compared to other groups. The diabetic conditions increase the production of ROS/RNS, which decreases the collagen synthesis [52]. An increase in the percentage of collagen density in the DAM indicated an increase in collagen synthesis.

**Table 8.** Cell counts considering the mean and standard deviation of collagen density.

| Group Treatment | Collagen Density (%) | | |
|---|---|---|---|
| | Day 3 | Day 7 | Day 14 |
| Normal (NC) | $53.0 \pm 1.4$ | $76.0 \pm 0.5$ | $88.0 \pm 0.9$ |
| Diabetic (DC) | $32.0 \pm 1.4$ | $44.0 \pm 0.2$ | $65.0 \pm 2.1$ |
| Alginate ointment (DA) | $43.0 \pm 2.8$ * | $65.0 \pm 3.5$ | $79.0 \pm 0.7$ |
| Alginate–mangosteen rind ointment (DAM) | $65.0 \pm 1.4$ * | $85.0 \pm 0.7$ | $91.0 \pm 0.7$ ** |

Asterisk notation in the table shows the result of the statistical test ($\alpha < 0.05$). * and ** notation shows insignificant differences with NC but significant differences with DC. * = $0.01 \leq \alpha < 0.05$; ** = $\alpha \leq 0.01$.

Achievement of the wound-healing process that occurred in DAM could be due to compounds possessing antioxidant activity from the combined mangosteen peel. Moreover, Kataria et al. (2014) reported that alginate is a good absorbent and gel formation agent that has homeostatic properties [15]. Alginate has been used in several types of wounds, including pressure wounds, diabetic wounds, and venous ulcers such as cavities and multiple bleeding sores [14]. In addition, alginate acts as an antioxidant that can capture free radicals and absorbs wound exudate.

## 5. Conclusions

The administration of a topical combination of alginate extracted from *S. ilicifolium* and mangosteen rind extract (*Garcinia mangostana*) in diabetic open wounds is able to accelerate the process of wound closure. This is evidenced with an increase in neutrophils,

macrophages, fibrocytes, fibroblasts, and collagen density, which enhance the process of re-epithelialization and the synthesis of collagen in the wound area. Alginate absorbs wound exudate, and xanthones are known to have antioxidant and anti-inflammatory activities. Xanthone is capable of capturing free radicals (ROS/RNS), which usually tend to increase in diabetics. Therefore, it can be concluded that the combination of alginate and mangosteen rind extract can improve the wound-healing process in diabetic mice.

**Author Contributions:** This paper is part of a master's degree thesis. Conceptualization, P.P. and D.W.; methodology, P.A.C.W. and S.A.H.; software, P.A.C.W.; validation, D.W., A.P., and S.A.H.; formal analysis, P.A.C.W., A.P., and M.V.; investigation, P.A.C.W.; resources, M.A.A.; data accuracy, P.A.C.W.; writing—original draft preparation, P.A.C.W.; writing—review and editing, P.P., K.A., and D.M.; visualization, P.A.C.W.; supervision, P.P. and D.W.; project administration, Z.N.I. and S.A.H.; funding acquisition, P.P. All authors have read and agreed to the published version of the manuscript.

**Funding:** This research was supported by the Innovation and Research Center, Airlangga University, Grant Number 1408/UN3/2019, and the APC was funded by the Innovation and Research Center, Airlangga University.

**Institutional Review Board Statement:** The study was conducted according to the guidelines of the Declaration of Helsinki, and approved by the Institutional Review Board (or Ethics Committee) of ANIMAL CARE AND USE COMMITTEE, Airlangga University (No.2.KE.049.04.2019, 4 April 2019).

**Informed Consent Statement:** Not applicable.

**Data Availability Statement:** The data presented in this study are available on request from the corresponding author, upon reasonable request.

**Acknowledgments:** The author would like to thank the Innovation and Research Center, Airlangga University for funding this research through the Mandat Research Grant, Airlangga University FY 2019 and Leonardo Gomez from the CNAP of the University of York for granting access to the SEC-MALLS system.

**Conflicts of Interest:** The author declares no conflict of interest.

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
