# Peer review of "Wound Healing and Antioxidant Evaluations of Alginate from Sargassum ilicifolium and Mangosteen Rind Combination Extracts on Diabetic Mice Model"

_applsci, doi:10.3390/app11104651_

Round 1

Reviewer 1 Report

Dear Editor,

The manuscript submitted by Pugar Arga Cristina Wulandari et al. and entitled: “Evaluation of Wound Healing and antioxidant activity of Alginate from Sargassum ilicifolium and Mangosteen Rind combination Extracts on Diabetic Mice Model” is an interesting work. This manuscript can be accepted after major revisions. Please to see my comments below.

Comments:

Authors should reformulate the title. It is too long and confused for reader understanding.

The introduction provides sufficient background and include relevant references. Maybe recent reviews or book chapters on alginate applications (food, pharmaceutical…) should be included in order to give potential bio-application of Alginate from Sargassum ilicifolium.

The methods are sufficiently described in the manuscript.

The research design and results are appropriate and clearly presented and discussed.

In results and discussion part, author should give more comparison and discussion with other publications using polysaccharides/polyphenol/xanthones applications. In result part, the data given could be more discussed.

General comment:

The quality of some figures must be improved in the revised manuscript.

Reviewer 2 Report

Dear Editor,

I am pleased to review the manuscript by Pugar Arga Cristina Wulandari et al: Evaluation of Wound Healing and antioxidant activity of Alginate from Sargassum ilicifolium and Mangosteen Rind combination Extracts on Diabetic Mice Model. After checking the novelty and creativity, I saw similar paper drafted by their group: A Novel Therapeutic effects of Sargassum ilicifolium Alginate and Okra ( Abelmoschus esculentus ) Pods extracts on Open wound healing process in Diabetic Mice, June 2020, Research Journal of Pharmacy and Technology 13(6):2764-2770, DOI: 10.5958/0974-360X.2020.00491.6. I believe these two papers may share a common group, which is ok for the publication.

Even there are some researches, this MS showed good design and the finding is interesting. After they administered a topical combination of alginate extracted from S. ilicifolium and mangosteen rind extract (Garcinia mangostana) in diabetic open wounds, it showed the ease the process of wound closure. Even more the combination of alginate and mangosteen rind extract can improve the wound healing process in diabetic mice. However, even though there are some merits, I suggest the final checking and some areas have to be improved the explanation or rewritten to meet the quality required or to make it more accurate and concise.

Author Response

Please see the attachment of manuscript after revising

Reviewer 3 Report

Authors report on wound healing and antioxidants activity of combination of extracts on a diabetic mice model. Extracts were suitably characterized and several valuable bioassays were performed, such as antioxidant activity, blood glucose level, wound width, number of neutrophils, macrophages, fibrocytes, fibroblasts etc. In general, the work was well performed and is presented in suitable fashion and would be as such interesting for broad scientific community. There is on minor issue: the order of data in the Abstract seems not to be well structured.

Round 2

Reviewer 1 Report

Authors revised manuscript according to reviewer's comments. Consequently this paper could be accepted in current form